# ACIDES: on-line monitoring of forward genetic screens for protein engineering

Takahiro Nemoto [1,2,3] ✉, Tommaso Ocari[1], Arthur Planul[1], Muge Tekinsoy[1], Emilia A. Zin [1], Deniz Dalkara [1] ✉ & Ulisse Ferrari [1] ✉

Forward genetic screens of mutated variants are a versatile strategy for protein engineering and investigation, which has been successfully applied to various studies like directed evolution (DE) and deep mutational scanning (DMS). While next-generation sequencing can track millions of variants during the screening rounds, the vast and noisy nature of the sequencing data impedes the estimation of the performance of individual variants. Here, we propose ACIDES that combines statistical inference and in-silico simulations to improve performance estimation in the library selection process by attributing accurate statistical scores to individual variants. We tested ACIDES first on a random-peptide-insertion experiment and then on multiple public datasets from DE and DMS studies. ACIDES allows experimentalists to reliably estimate variant performance on the fly and can aid protein engineering and research pipelines in a range of applications, including gene therapy.

Directed evolution (DE)[1–3] is a versatile protein engineering strategy to conceive and optimize proteins like enzymes[4–6], antibodies[7,8] or viral vectors for gene therapy[9–15], culminating in the Nobel Prize in Chemistry 2018[16]. DE starts from a massive library of random mutants, screens it against a given task over multiple rounds and searches for the variants with the highest performance. As the iteration continues, the best performing variants get enriched and emerge from the bulk, while ineffective ones are instead weeded out. Nowadays, we can rely on next generation sequencing (NGS)[17,18] to sample millions of variants within the library and monitor their concentrations over multiple rounds or time-points. In this approach, the enrichment of the screened variants is measured to rank the variants depending on their performance. In a similar flavor, Deep mutational scanning (DMS) experiments[19–21] combine extensive mutagenesis with NGS to study the properties of proteins[22–32], viruses[33,34], promotors[35,36], small nucleolar RNA[37], tRNA[38,39] or of amino-acid chains. It uses similar techniques to DE and requires similar analysis. Both methods are based on forward genetic screens, and the approach presented in this article can be applied to these fundamental techniques, focusing on their common issues and needs.

The analysis of NGS data of multiple selection rounds and/or multiple replicate experiments presents several difficulties. First, variants need to be robustly scored based on their enrichment rates, so-called selectivities[40,41]. This task is complicated by the large noise in the NGS counts introduced by, for example, polymerase chain reaction (PCR) amplification or bacterial cloning, during amplicon preparations[42–44]. This noise needs to be taken into account in the analysis. Second, in order to rank the variants and to identify the best performing ones, the score should come with a precise estimation of its statistical error. As a consequence of the noise in the counts, some irrelevant variants might appear to be highly enriched (winner's curse). This would be anticipated if properly estimated credibility scores are available. Third, when running DE over multiple rounds, it is hard to know when to end the experiment: performing too few rounds could lead to selection of weak variants, not representative of their true ranking. On the other hand, performing too many rounds is costly, time-consuming and even ethically questionable when working with in-vivo selections[14,45]. Similarly, it would be useful to understand the best NGS depth for a given experiment, as deepening the NGS by increasing reads results in better data, but adds an extra expense to the experiment.

[1]Institut de la Vision, Sorbonne Université, INSERM, CNRS, 17 rue Moreau, 75012 Paris, France. [2]Graduate School of Informatics, Kyoto University, Yoshida Honmachi, Sakyo-ku, Kyoto 606-8501, Japan. [3]Premium Research Institute for Human Metaverse Medicine (WPI-PRIMe), Osaka University, Suita, Osaka 565-0871, Japan. ✉e-mail: nemoto.takahiro.prime@osaka-u.ac.jp; deniz.dalkara@inserm.fr; ulisse.ferrari@inserm.fr

In order to account for these issues and needs, we present ACIDES, Accurate Confidence Intervals for Directed Evolution Scores, a computational method to empower the analysis of DE and DMS experiments. We focus on screening experiments on highly diverse libraries where massive NGS data are collected over multiple rounds, multiple time-points and/or multiple replicates (Fig. 1a). Our goal is to develop a method to extract maximal information from noisy NGS data, and allows for scoring and ranking variants with accurate statistical confidence scores. Our approach can be applied to various types of experiments. These include in-vivo DE[13,14,46], DMS of phage-display[24,40,47], yeast two-hybrid assays[24], small nucleolar RNA studies[37], mRNA display[23] as well as cell-based DMS experiments[26,29,31,38,39,48], among others. It is

possible to apply ACIDES either *a posteriori* over data collected previously, or along the course of the experiment as soon as the NGS data become available. The latter strategy allows for monitoring the selection convergence on the fly, and to understand when the experiment can be ended. In this way, ACIDES can be integrated into protein engineering pipelines as well as studies of protein function using mutagenesis. The tutorial for using ACIDES, along with an executable code in Python, can be found in https://github.com/nemoto-lab/ACIDES/.

## Results

The first step of ACIDES estimates the selectivity of each individual variant present in the dataset (Fig. 1b) and its 95% confidence interval

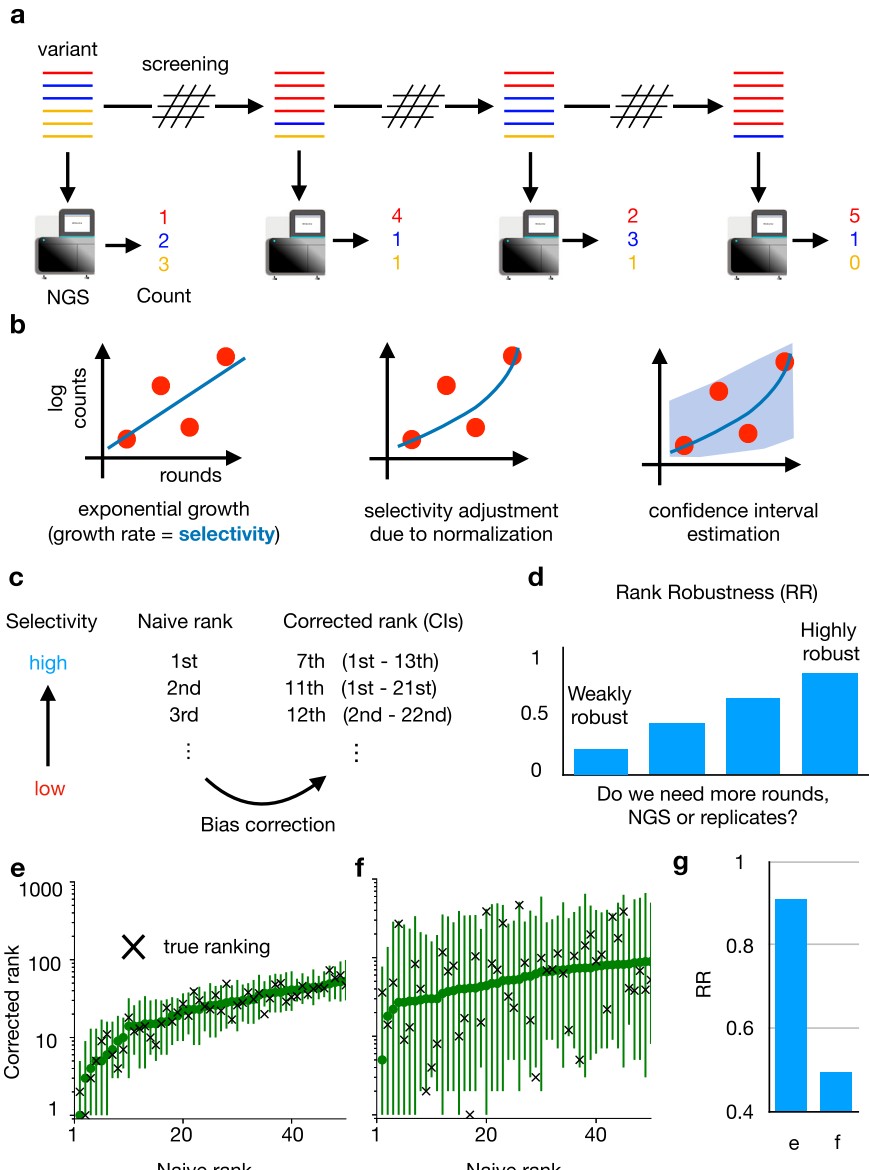

**Fig. 1 | ACIDES framework. a** We consider directed evolution (DE) experiments, where protein variants are screened over multiple rounds, and massive NGS datasets are collected. **b** From the obtained count data, we estimate a score (selectivity) for each variant. The higher the score, the more the variant is adapted for the screening task. Each score is computed using a maximum likelihood estimation (Methods). 95%-confidence interval (CI) is defined as two standard deviations from the mean. **c** Sorting the scores of all variants in descending order, we obtain a variant rank (naive rank). Due to statistical errors in the scores, the obtained rank is biased in general. To correct for this, using *in-silico* simulations based on the CIs of the scores, we re-estimate the rank with 95%-CI (corrected rank). **d** From the obtained corrected rank, we compute Rank Robustness (RR). RR

represents the percentage of the top 50 variants identified in the naive rank that also appear in the top 50 of the corrected rank. **e**, **f** Examples of rank graphs for two synthetic datasets with different depths of NGS (per round) and numbers of unique variants (respectively, E: $10^7$, $5 \times 10^4$; F: $10^6$ and $10^6$). The medians (the green circles) and the 95%-CI (error bars) of the corrected ranks are estimated from 3000 bootstrap samples (Methods), using the 2.5th, 50th, and 97.5th percentiles. The true rank is shown as red crosses. In both cases, most red crosses are within the 95%-CI. **g** RR for the two synthetic datasets. Note that RR multiplied by 50 (E:-45.3; F:-24.6) roughly provides the number of the correct top-50 sequences, which are 46 and 23, respectively. (See Figs. S3 and S4 for more systematic comparison. Source data are provided as a Source Data file.

**Table 1 | Next generation sequencing datasets of directed evolution experiments**

| Label | Experiment | Target | Time-points | Reads/round | # vars | Avg. count | Accuracy | Spread | Ref. |
|-------|-----------|--------|-------------|-------------|--------|-----------|----------|--------|------|
| A | Phage display | BRCA1 | T0 → T5 | 8.3 M | 35 k | 240 | 3.06 | 0.83 | Starita 2015[24] |
| B | Yeast two-hybrid | BRCA1 | T0 → T3 | 13.5 M | 27 k | 490 | 1.83 | 0.74 | Starita 2015[24] |
| C | Phage display | Ab IgH | T1 → T3 | 0.1 M | 29 k | 3.7 | 0.95 | 0.96 | Boyer 2016[47] |
| D | Phage display | hYAP65 WW | T0 → T3 | 5 M | 470 k | 11 | 1.40 | 0.77 | Araya 2012[40] |
| E | Dog eye DE | AAV2-7mer | T0 → T5 | 17 M | 5 k | 3360 | 0.44 | 1.88 | Byrne 2018[46] |
| F | Yeast growth | U3 snoRNA | T0 → T4 | 8 M | 24 k | 352 | 1.22 | 0.54 | Puchta 2016[37] |
| G | Murine lung DE | AAV2-7mer | T1 → T5 | 6.2 M | 0.5 k | 1000 | 0.34 | 2.00 | Korbelin 2016[13] |

List and properties of experiments considered in this study. First column introduces dataset label and corresponds to the panels of Fig. 2. *Reads/round* corresponds to the average NGS counts per time points. *# vars* is the number of unique variants that is detected in the NGS at least once during whole experiments. *Avg. count* is the mean NGS count over all variants and round. *Accuracy* is the inverse of average errors predicted by the model for the estimated scores of the top 1000 variants, normalized by the highest score and divided by 10. *Spread* is the difference between the estimated scores of the highest and 1000th highest variants, normalized by the highest score.

(95%-CI). In this study the term selectivity means the rate at which each variant increases its concentration with respect to the others. More precisely, we assume an exponential growth as $\rho^i_{t+\Delta t} \sim \rho^i_t \exp(a^i \Delta t)$, where $\rho^i_t$ is the concentration of variant $i$ at time $t$, and $a^i$ is its selectivity. Compared with previous methods[19–22,24,25,35–37,40,41,49], our approach combines a robust inference framework (maximum likelihood estimation) with a better quantification of the NGS sampling noise[42–44]. For this scope, our approach benefits from a negative binomial distribution that has been intensively used in differential gene expression analysis[50–52]. In the negative binomial distribution, the variance of the noise is overdispersed and grows as $\lambda + \lambda^{2-\alpha}/\beta$[53] (Fig. S1). Here $\lambda$ is the expected mean count, and $\alpha, \beta$ are parameters to be inferred (Methods). Using novel data from a plasmid library, we observed that our negative binomial model realizes a 50- to 70-fold improvement over the Poisson model in the predictive ability of the NGS sampling noise (Fig. S1). The second step of ACIDES uses the estimation of the selectivities and their statistical errors to rank the variants. The rank obtained by sorting the selectivities in descending order (naive rank) is biased due to statistical fluctuations of the selectivities. We correct this bias using *in-silico* simulations (Fig. 1c). The third and last step of ACIDES uses simulations to quantify a Rank Robustness (RR), a measure of the quality of the selection convergence (Fig. 1d). Specifically, RR is the ratio at which the top-50 variants in the naive rank are correctly identified (Methods). RR ranges from 0 to 1: a low value points out that the variants have not been selected enough, and therefore calls for the necessity to perform more rounds, deeper NGS sampling or possibly more replicates. Conversely, a large value confirms that the selection has properly converged, and suggests that the experiment can be ended without performing additional experimental steps.

Before focusing on experimental data, we apply ACIDES to two synthetic datasets (Methods) describing two opposite scenarios (See Figs. S3 and S4 for more systematic comparison): data-rich case (more NGS reads with fewer unique variants) and data-poor case (less NGS reads with more considered variants). These datasets are generated using the negative binomial distribution, thus serving as an idealized testing ground. In the data-rich case, we first verify that our method reaches high performance in recovering the ground-truth values of the selectivities ($R^2 \simeq 0.92$, Fig. S3) in a teacher-student setting. In this first case, selection convergence is reached and the different variants can be robustly ranked (Fig. 1e). In the data-poor case, instead, CI-bars are large and the ranking is uncertain (Fig. 1f). Consistently, the estimated RRs are high and low for, respectively, the data-rich and -poor examples (Fig. 1g). Note that, once multiplied by 50, RR roughly provides the number of the correct top-50 variants in both cases (caption of Fig. 1g). Furthermore, we observe that most true rank values (red crosses) fall within the 95%-CI in both examples. These observations show that our approach can quantify statistical errors even in the data-poor regime (See Fig. S4 for more systematic comparison).

## Analysis of screening experiments with multiple time points

In order to showcase ACIDES, we apply it to several screening datasets with multiple time points, where various proteins (and one RNA molecule) are screened using different experimental techniques (Table 1). Specifically, we consider three phage-display screening experiments targeting different proteins, such as the breast cancer type 1 susceptibility protein (BRCA1) for Data-A, human yes-associated protein 65 (hYAP65) for Data-D and immunoglobulin heavy chain (IgH) for Data-C, two in-vivo DEs of adeno-associated virus type 2 (AAV2) vectors targeting canine eyes for Data-E and murine lungs for Data-G, a multiplexed yeast two-hybrid assay targeting BRCA1 for Data-B and a yeast competitive growth screen measuring the fitness of mutant U3 gene for Data-F. For each of these experiments, we rank variants (naive rank) and compute the confidence interval of their ranks (corrected rank in Fig. 2a–g). The degree of convergence of the selection is quantified by RR (Fig. 2h). When technical replicates are available (Data-A and Data-B), we compute RR over all of them and obtained consistent results (shown by the small error-bars in Fig. 2h).

To gain deeper insight into RR, we introduce the following two metrics: (i) Accuracy: this measures the accuracy of variant performance measurements, computed based on the inverse of average errors predicted by the model for the estimated scores (Table 1), and (ii) Spread: the extent to which the variants are intrinsically different in their performance for the task, calculated based on the difference between the estimated scores of the highest-scoring variant and the variant ranked 1000th (Table 1). The larger these quantities are, the easier it becomes for us to distinguish the best-performing variants, thus resulting in a higher RR. In Data-A and Data-B, Accuracy is the highest, while the Spread is also relatively large. This is consistent with the high RR (RR > 0.8). In Data-C and Data-D, the value of RR ranges between 0.6 and 0.8. The experimental techniques used in these datasets are similar to those in Data-A and Data-B, which could be related to similar Spreads between them. But the average NGS counts (NGS depths relative to the number of sequences, Table 1) and Accuracies are smaller in Data-C and Data-D, resulting in lower RRs. In Data-E, Data-F, and Data-G, RR values are low (~0.5), but the reasons for these low values are different. Data-E and Data-G (in-vivo DE experiments using AAV) suffer from very low Accuracies in their experiments, even though they have large Spreads. On the other hand, Data-F has a relatively high Accuracy, but Spread is the narrowest, meaning that their variants perform similarly in their experiment. These two factors counterbalance each other, resulting in similar RR values for these datasets.

In datasets with low RRs, some variants seem to be more adapted to the screening task than the others, but the difference between their scores is marginal compared with their statistical errors. This means that we cannot distinguish if the obtained variants are selected because of their ability to perform the task (fitness) or just there due to noise. In these cases, experimentalists have two possibilities: (i) based on the noisy identified variants, perform further tests in addition to

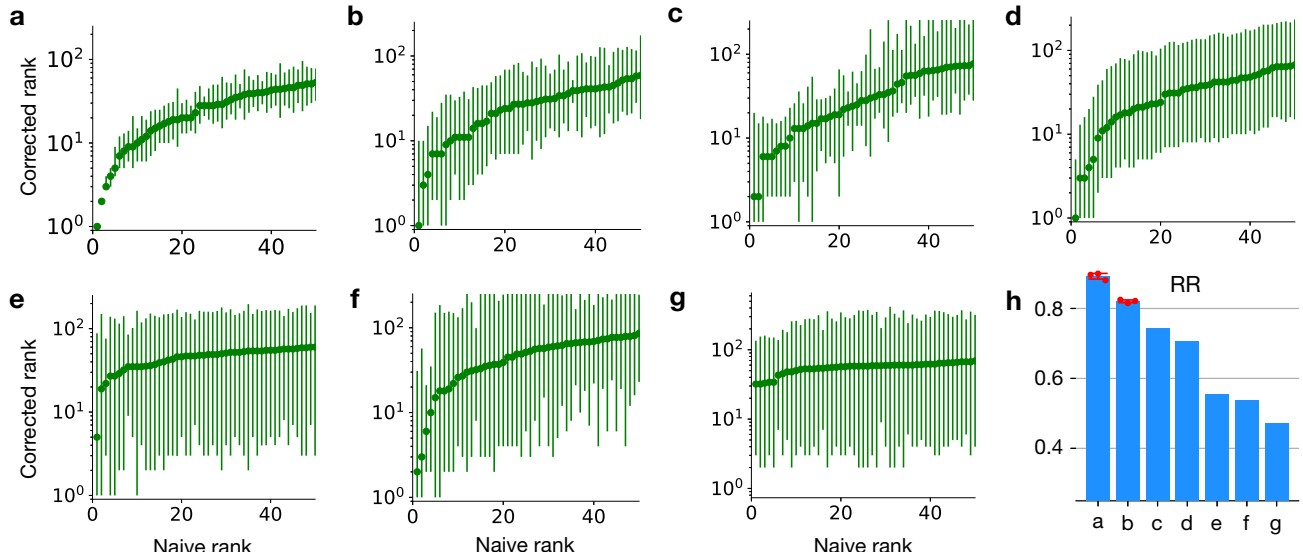

**Fig. 2 | Rank graph for various experimental datasets. a–g** The panel labels **a–g** correspond to the experiments listed in Table 1. The corrected ranks are shown with 95%-CI, estimated using 3000 bootstrap samples (Methods). **h** Rank robustness (RR) for each experiment. For Data-A and -B, the mean and standard deviation estimated from three technical replicates are shown. Source data are provided as a Source Data file.

DE[13,14], as for example, study infective ability of viral vectors using single-cell RNA-seq[54]. Or (ii) increase the quality of the datasets, by performing further selection rounds, increasing NGS depths, or replicating the experiments under the same conditions. This second possibility is explored in the next section. Overall our rank-analysis of the different experiments shows how our approach can provide an overview of the selection convergence, informing about the state of the experiment and eventually pointing out the necessity of more experimental efforts.

### Integration into the experimental pipeline

Noise in experimental data can be reduced by performing additional selection rounds involving experiments, but in general these are expensive, time-consuming and, in case of experiments involving animal use, ethically problematic[45]. For these reasons, it is important to choose accurately the number of rounds and the NGS depth. For this scope, ACIDES can be integrated into experimental pipelines to obtain an overview on how RR depends on these factors. This is to help experimentalists make informed decisions about additional experimental efforts.

ACIDES can estimate RR after each selection round (or any time new data become available). This allows us to examine the data's behavior and to quantify the degree of convergence in terms of the selection rounds. Similarly, for each round, ACIDES can be run on downsampled NGS data to compute RR with smaller NGS depth (Methods). Using these two techniques, we monitor the need for more selection rounds or deeper NGS: a slow increase of RR (or no change in RR) upon improving data-quality implies that convergence is reached and suggests that the experiment can be ended. If, on the other hand, RR increases rapidly when improving the rounds and/or NGS depth, it is probably worth making further experimental efforts.

In order to showcase our approach, we study how RR depends on the number of screening rounds and NGS depth in previous experiments. We start by measuring RR in Data-A for different NGS depths. 95%-CI on corrected ranks gets larger as the NGS depth becomes smaller (Fig. 3a). At 1% NGS depth, the variant ordering seems largely unreliable: RR is smaller than 0.5 (Fig. 3b). Importantly, RR does not decrease smoothly as the NGS depth decreases, but it remains roughly constant at the beginning, and falls only at a very small NGS depth. This result suggests that the actual NGS depth of this experiment largely

exceeds what was necessary (10% of the depth would have been sufficient). Next, we quantify how RR depends on both the number of performed rounds and NGS depth (Fig. 3c). RR grows from 0.28 (3 performed rounds with 1% NGS depth) to 0.88 (6 performed rounds with 100% NGS depth). Saturation of RR seems to be observed for RR > 0.7, which corresponds to 5 performed rounds with the NGS depth larger than 20%, or 4 performed rounds with the NGS depth larger than 40%. This again indicates that the experiment could have been stopped earlier (less rounds and/or lower sequence coverage) without much affecting the outcome. Note that different datasets show different behaviors. For Data-F more selection rounds with a higher number of NGS reads is expected to improve RR, while for Data-B they seem to have just reached the saturation point (Fig. 3d).

Overall these results show how our approach can be implemented along experimental pipelines. By estimating RR while collecting new data, we can understand if we should continue/stop adding more rounds or increasing NGS depth. This could avoid unnecessary, costly and time-consuming experimental efforts. Similar analyses can be done on the number of replicate experiments (Fig. S6).

### Comparison with previous work

We start by comparing the performance of ACIDES with Enrich2, the state-of-the-art for estimating variant scores (selectivities) in multiple time-points experiments[41]. Enrich2 is based on a weighted linear fitting of the log-count change over the course of rounds, and the first step of ACIDES should be seen as an upgrade for this fitting. Both algorithms predict standard statistical errors associated with the estimated scores (Methods) without using any replicate experimental data. In this comparison, our focus is on the accuracy of these errors. To test the accuracy, we leverage replicate datasets. We first investigate if the scores associated with low predicted errors in each method are consistent over replicates. For this, we plot the scores with low predicted errors obtained from one replicate against those obtained from the other (Fig. 4a, b). The correlation between replicates is estimated using the coefficient of determination ($R^2$). The correlation quantifies the quality of the predicted errors, as higher (or lower) correlations imply that the estimated scores are more (or less) robust, as attested by the low predicted errors. The figure shows that ACIDES outperforms Enrich2. Next, we test how the comparison depends on the data size. To this goal, we systematically select a set of variants based on the

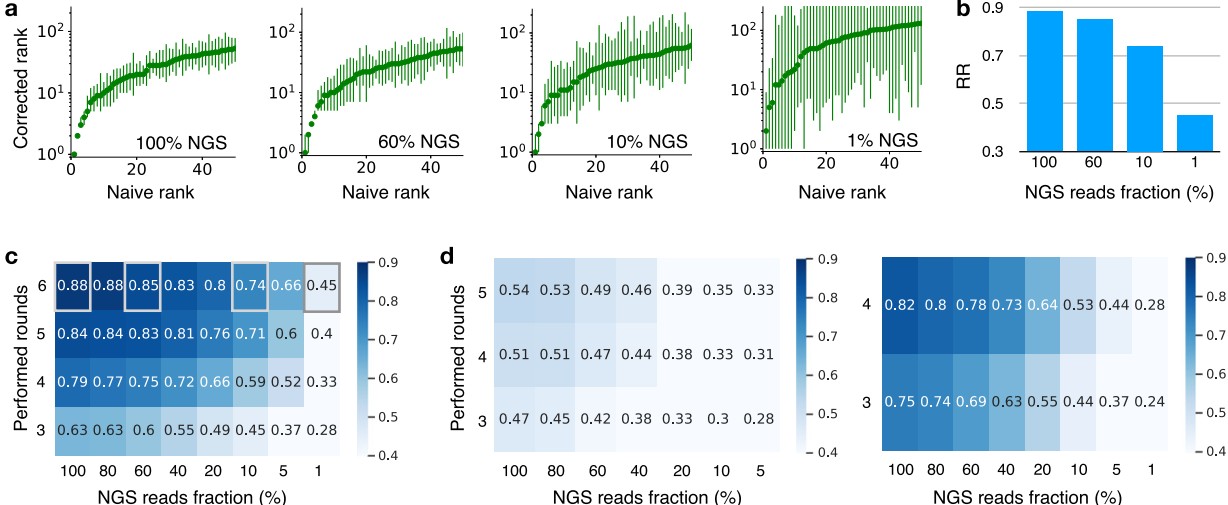

**Fig. 3 | How the rank robustness depends on the experimental protocol. a** The corrected ranks with 95%-CI for different NGS depths in Data-A (Table 1). 95%-CI is estimated using 3000 bootstrap samples (Methods). Different NGS-depth data are generated using downsampling (Methods). $x\%$ means the dataset where the number of NGS reads per round is reduced to $x\%$ (100% is the original dataset). **b** RR for the rank graphs in the panel **a**. Note that RR is higher than 0.7 even with the 10% NGS-depth. **c** The heat map showing RR for various NGS depths and performed rounds in Data-A. RR is larger than 0.7 for the data with (i) the 4 performed rounds with the NGS depth larger than or equal to 40% or with (ii) the 5 performed rounds with the NGS depth larger than or equal to 20%. This indicates that the data quality was already high with less experimental efforts. The four grey squares correspond to the four rank graphs in panel **a**, respectively. **d** The same graphs as the panel **c**, but for different datasets. Data-F is used in the left panel, where RR is low and more NGS and/or screening rounds would be useful. Data-B is used in the right panel, where RR takes high values and seems to saturate in NGS depths. Further experimental efforts would probably not be necessary in this dataset. Source data are provided as a Source Data file.

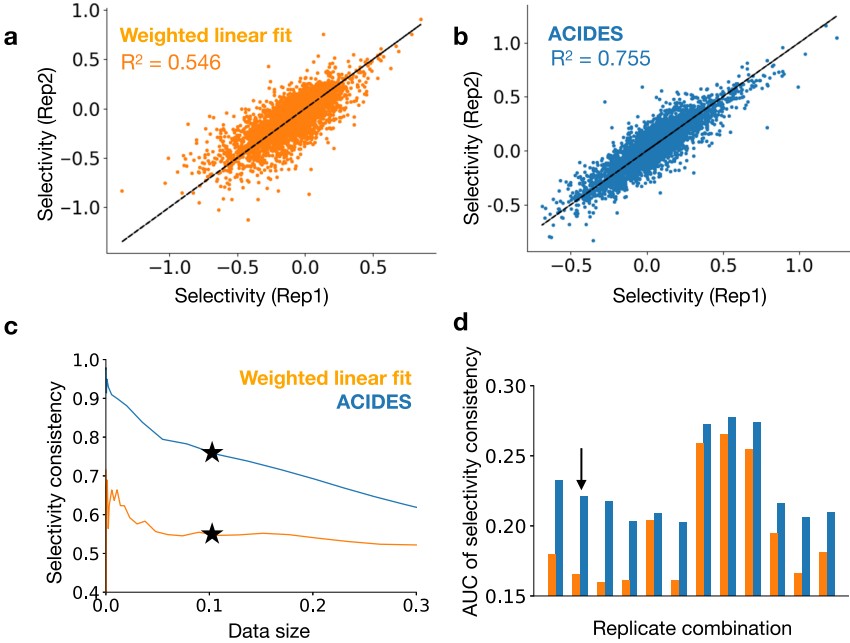

**Fig. 4 | Comparison of ACIDES with the state of the art: multiple time-point experiments. a**, **b** Using technical replicates in Data-A, we compare ACIDES with a weighted linear least squares method (Enrich2 (**a**))[41]. For both methods (Enrich2 (**a**) and ACIDES (**b**)), the inferred selectivities from one replicate are plotted against the selectivities in the other replicate. The coefficient of determination ($R^2$), which quantifies the consistency between two replicates, is also shown. **c** We next examine how the comparison in panels **a** and **b** depend on data size. We consider a set of variants in which the predicted statistical errors (Methods) are smaller than a given threshold. Varying this threshold, sets of variants are systematically selected, where larger/smaller sets include variants with larger/smaller estimated statistical errors. For each set, we estimate $R^2$ between two replicates, and plot it as a function of the set size. The panels **a** and **b** correspond to the stars ★ in C (data size 0.11). **d** In order to test both methods more systematically, we perform the same analysis (as those in panels **a**–**c**) for all possible 12 combinations of technical replicates in Data-A and Data-B. We define the area under curve of $R^2$ (in the panel **c**) and plot it for these combinations (**d**). Our method systematically outperforms the weighted linear fitting method. The replicate combination used for panels **a**–**c**) is indicated by the arrow in panel **d**. Source data are provided as a Source Data file.

magnitude of predicted errors. (Smaller/larger sets include variants with lower/higher predicted statistical errors.) For each set, we measure the correlation between two replicates as in (Fig. 4a, b), and plot it as a function of the set size (Fig. 4c). We observe ACIDES's correlation becomes more dominant as the set size decreases, consistently suggesting the better quality of the predicted statistical errors. In order to generalize these results, we perform the comparison for all possible 12 pairs of technical replicates in Data-A and Data-B (Table 1). In all cases our approach outperforms the competitor (Fig. 4d). We also conduct additional tests to quantify the consistency of the predicted statistical errors (Fig. S7) and demonstrate the higher capacity of ACIDES to recover the ground truth rank in several scenarios based on synthetic data (Fig. S8).

Next, we tested the performance of ACIDES on 12 datasets[23,26,29,31,38,39] (taken from a test performed in ref. 48) that have only two time-points, but multiple replicates. We compare ACIDES with several different algorithms, including Enrich2[41] and DiMSum[48], which is another state-of-the-art algorithm specilized to two time-point datasets with multiple replicates. In this comparison, we employ the cross-validation techniques used in[48] to calculate a z-scores of enrichment estimations for each variant (see Method for more details). Better the algorithm is, the closer the distribution of z-scores are to a standard normal distribution. In order to compare the different algorithms, we calculate: (i) the inverse of the standard deviation of the z-scores (as in ref. 48) and (ii) the $R^2$ scores, a measure of the shape difference between the z-score distribution function and the standard normal distribution function, reflecting higher-order statistics beyond the inverse standard deviation (i). This score is derived from a quantile-quantile plot and its comparison to the $y = x$ line (see Method). In both cases, values closer to 1 indicates good performance of the algorithm. In the first test using (i) (the red box plots in Fig. 5), ACIDES and DiMSum show similar performance and they both overperfom Enrich2. In the second test using (ii) (the blue box plots in Fig. 5), ACIDES slightly outperforms DiMSum and behaves better than the other algorithms.

## Discussion

In this work, we have presented ACIDES, a method to quantify DE and DMS selectivities (fitness), rank variants with accurate credibility scores and measure the degree of experimental convergence. ACIDES can be used on the fly to offer an overview of the progress of selection experiments, which would help experimentalists with making informed decisions on whether new experimental efforts are needed. In this way, ACIDES can save significant experimental time and resources. We have applied ACIDES to several DE and DMS datasets where a number of different target proteins and RNA molecules have been screened using different experimental protocols. The heterogeneity of these datasets shows that ACIDES is a method of general use, applicable to many different experiments.

The first step of ACIDES estimates the score (selectivity) of each observed variant. This is a necessary step, and several alternative methods have been proposed in the past. In many applications with multiple time-points, such scores are computed as the variant enrichment that is defined as the logarithmic ratio between the variant frequencies in the last and second to last round[13] or between the last and first round[14,19,20,22,35,49]. These approaches thus make use of data from only two rounds and disregard all the others. For this reason, this strategy is suboptimal and may lead to noisy score estimations. A more sophisticated approach that uses all the data consists in inferring the slope of a linear line fitted to the log-frequencies of variants over all the screening rounds/time points[24,25,36,40]. This method gives the same importance to log-frequencies in all the rounds. Yet as variant counts in the first rounds are typically small and noisy, assuming the same weight on them could result in an overfitting. To fix this effect, Enrich2[41] uses the variance of the count data - estimated via a Poisson

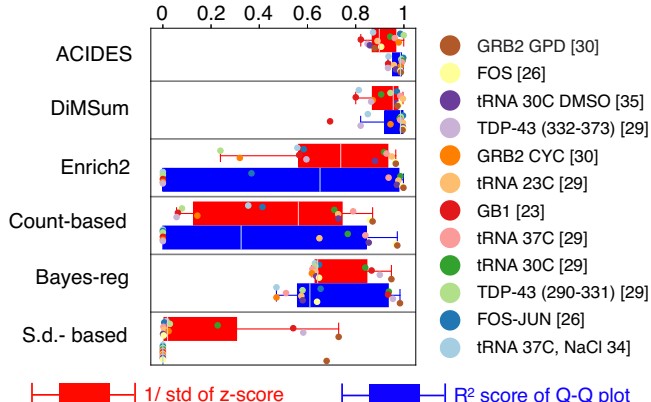

**Fig. 5 | Comparison of ACIDES with the state of the art: two time-point experiments.** We test the accuracy of error predictions from 6 different models using 12 sets of two-time point experimental data, each with replicates. The z-score, defined as the variant-score difference between replicates divided by the predicted errors, should ideally follow the standard normal distribution if the model's predictions are accurate. To evaluate how closely the z-score distribution matches the standard normal distribution, we first compute the inverse of the standard deviation of the z-score distributions[48] (see the main text and Methods). These results are shown in red box plots, where values closer to 1 indicate better fits. We next compute a metric that encompasses broader statistical aspects than just the standard deviation to assess the similarity between the z-score distribution and the standard normal distribution (details can be found in the main text and Methods). These results are displayed in blue box plots, and as before, values closer to 1 indicate better model fits. The box extends from the first quartile to the third quartile of the data, with a line at the median. The whiskers extend from the box by 1.5x the inter-quartile range. Source data are provided as a Source Data file.

distribution assumption - as the weights in a linear least squares fitting. ACIDES' first step comes with a three-fold improvement over this last approach. First, instead of relying on the linear least squares fitting, we estimate the score by log-likelihood maximization. A major improvement happens for variants whose log-frequencies do not grow linearly with the rounds, and a simple linear weighted fit may struggle in identifying the correct slope. This is particularly visible in the bulk variants with intermediate scores (Fig. 4 a, b). Secondly, instead of a simple exponential growth of the counts, we included a softmax non-linear function (Methods), where the denominator is inferred from data[55]. This change improves the score estimation when the wildtype (if any) and/or few variants have a large fraction of the total counts and bend the exponential growth of the log-frequencies. Lastly, ACIDES uses a negative binomial distribution to model the count variability[50–53]. This distribution accounts for the large dispersion of next-generation sequencing data[42–44] far better than the Poisson distribution (Fig. S1). Additionally, the negative binomial loss in the likelihood maximization allows us to better estimate statistical errors for the inferred scores. Thanks to all these improvements, our approach realizes a more robust and accurate estimation of the variant scores and outperforms the previous method (Fig. 4).

In Fig. 5, we compared ACIDES with DiMSum[48] a recently developed algorithm for analyzing DMS experiments. DiMSum is tailored specifically for experiments involving two time-points with multiple replicates, while ACIDES is more general as it can be applied to multiple time-points cases, even without replicate data. DiMSum directly models the errors of each variant enrichment, by adjusting Poisson-based errors to account for overdispersion. On the other hand, ACIDES models the fluctuations of the counts themselves, using an exponential model (1) and a negative binomial log-likelihood (2). Consequently, it can be used to compute the statistics of any quantity as a function of counts (see Method). This difference also appears in the statistics of z-score beyond the standard deviation (as shown in the blue box plot

in Fig. 5). Since $R^2$ score reflects the extent of similarity to the normal distribution, going beyond just comparing standard deviations, it indicates that, for these higher-order statistics, ACIDES slightly outperforms DiMSum and remains better than the other previous methods.

In the case of noisy data, the estimated scores of variants come with statistical errors. This means that the rank obtained from the scores (naive rank in our figures) is in general biased: top-ranked variants are overvalued, and vice-versa. This simple statistical effect was not taken into account in previous analyses related to DE and DMS experiments. The second step of ACIDES uses a bootstrap method to account for the bias and recover both the corrected rank and its 95%-CI. The deviation between this 95%-CI and the naive rank shows us how much we can trust the naive rank. To quantify it, as a third step of ACIDES, we introduce RR that describes how many of the top-50 variants in the naive rank are correct. RR measures how stable and robust are the ranks of the variant selectivities. As such, it quantifies the degree of convergence of the experimental selection, providing an insightful overview of the state of the experiment.

Although ACIDES demonstrates advantages over the other methods, it has several limitations that may be addressed in the future. First of all, ACIDES does not account for changes in the selection pressure over rounds. This can potentially be included, but has not been done here, as the selection pressure is constant in most datasets we analyzed in this article. Second, ACIDES uses a negative binomial model to describe the dispersion of count data by assuming that the count variance depends only on the frequency of the variant. Although this assumption proves useful to describe NGS count errors (Fig. S1) and is used elsewhere[53], it is possible that dispersions induced by a sequence-dependent procedure, such as error-prone PCR[14,46,56] (see later). Third, statistical errors due to the replicates that do not share the same initial library cannot be described by ACIDES, provided that the model is only trained on a single series of screening rounds. To account for this, we would need a framework that generalizes ACIDES for different sources of variability. Finally, the combination of Fluorescence Activated Cell Sorting (FACS)[57,58] with deep mutational scanning has recently been gathering attention[33,59,60], for example, in the study of the SARS-CoV-2 receptor binding domain[33]. Although the background model of ACIDES (1) needs slight modifications, these could enable us to use the same negative-binomial noise model and the log-likelihood maximization method. We defer this interesting line of research to future developments.

In its current version, ACIDES can not account for additional rounds of mutagenesis performed between the selection rounds. When this happens, ACIDES will treat the newly generated variants as if they had zero counts at the beginning and the scores for these new variants may be slightly underestimated. Note that, as error rates of mutagenesis are in general less than a few percent per position, in most cases, this has little practical impact on variants that already existed. Overall, we need to be cautious about any interesting variants that begin to emerge after the mutagenesis round. In our data, Data-E includes an error-prone PCR after the third round of selections. However, we have confirmed that all the top-50 variants observed in Data-E were already present from the start of the experiment.

Finally, using machine learning techniques, several studies have aimed at estimating selectivities from the amino-acid sequences of variants. Most of these methods rely on supervised algorithms, which are trained to predict the selectivity (output) from the sequence of a variant (input)[56,61–68]. Because the performance of these methods depends on how the selectivity is estimated from data, ACIDES can potentially be incorporated in their pipelines to improve the overall performance. We leave such analysis for future developments. Other methods use instead unsupervised approaches to predict selectivities from the sequences of variants[55,69–72]. Even if these methods do not use

any sequence scores for their training, they often use it to validate and/or test the model. Our approach would therefore be useful also in these cases.

## Methods

### Library preparation for Fig. S1

To demonstrate that our negative binomial likelihood approach outperforms the Poisson counterpart, we conducted the following experiment: We inserted random 21 nucleotide oligomers into a Rep-Cap plasmid containing adenoassociated virus 2 (AAV2) cap gene using previously described methods[73]. For this, the library was created from the pAV2 plasmid from the ATCC plasmid bank (American Type Culture Collection). The primers used below were synthesized by Invitrogen or Eurofins Genomics. We first inserted the following nucleotide sequence between amino acids 587 and 588 of the cap2 gene in the pAV2 plasmid: GCGGCCGCCTAGGCG. In order to insert the 21 nucleotides into the cap2 gene, an amplification of the sequence was then carried out using overlapping primers on the pAV2+AscI vector CGCAGCCATCGACGTCAGACGCGGAAGCTTCGATCAACTACGCAGAC and CTTGTGTGTTGACA TCTGCGGT AGCTGCTTGGCGCGCCNBNN BNNBNNBNNBNNBNNBNGCCGCGTTGCCTCTCTGGAGGTTGGTAGAT AC. We used the PrimeStar Mix®kit (Takara) and the following PCR program: 5 min at 98 °C, then, 30 cycles (10 sec at 98 °C; 5s at 55 °C; 15 sec at 72 °C) followed by 5min at 62 °C. The insert was then recovered on a 0.8% agarose gel using the NucleoSpin®Gel and PCR Clean-up kit (Macherey-Nagel). The plasmid library obtained was deep sequenced using Illumina NextSeq 500 following the generation of amplicons corresponding to the 7mer insertion region. Since the 21 nucleotides are randomly and independently generated, we can use a position-weight matrix model to predict the frequency of each variant in the sample. Based on this property, the performance of the two models are examined as shown in Fig. S1.

### Model

We propose ACIDES for analyzing selection data in DE and DMS. The mathematical model is described in detail below. For a given series of samples over screening rounds, we perform NGS and denote by $n_t^i$ the obtained count of the $i$-th variant ($i = 1, 2, \ldots, M$) at round (time-point) $t \in T$. We denote by $N_t$ the total count $N_t = \sum_i n_t^i$ at $t$. For each sample, we define $\rho_t^i$ as the expected value of frequency of the $i$-th variant at $t$. (Note that "expected" means that $\rho_t^i$ itself does not fluctuate due to the noise in the experiment.) For each variant, an initial frequency $\rho_0^i$ and a growth rate $a^i$ are assigned, by which the expected frequency is computed as

$$\rho_{t+\Delta t}^i = C_t \rho_t^i \exp(a^i \Delta t), \tag{1}$$

where $\Delta t$ is the round- (or time-) difference between two consecutive NGSs. $C_t$ is a normalization constant, defined as $C_t = 1/\sum_i[\rho_t^i \exp(a^i \Delta t)]$. We call this model (1) an exponential model.

We use a negative binomial distribution $\text{NB}(n_t^i|\lambda, r)$ with two parameters $\lambda$ and $r$ to model the noise distribution of counts $n_t^i$. Here $\lambda$ is the expected value of count $n_t^i$ given as $N_t \rho_t^i$, while $r$ is the dispersion parameter that describes the deviation of the negative binomial distribution from the Poisson distribution. (The negative binomial distribution is a generalization of the Poisson distribution with a variance equal to $\lambda(1 + \lambda/r)$: the Poisson distribution is recovered in the large $r$ limit.) Here, based on Fig. S1 and ref. 53, we assume $r$ is a power-law function of $\lambda$: $r(\lambda) = \beta \lambda^\alpha$ (with $\alpha, \beta > 0$), where $\alpha$ and $\beta$ are parameters that are common for all the variants in the experiment. (The variance is thus $\lambda + \lambda^{2-\alpha}/\beta$.) Model parameters $\alpha, \beta$ as well as $\rho_0^i, a^i$ ($i = 1, 2, \ldots, M$) are inferred from the count data $n_t^i$ ($i = 1, 2, \ldots, M, t \in T$) by maximizing the

following likelihood function:

$$L\left(\alpha,\beta,(\rho_0^i)_{i=1}^M,(a^i)_{i=1}^M\right) = \prod_{i,t} \text{NB}\left(n_t^i | \rho_t^i N_t, \beta(\rho_t^i N_t)^\alpha\right). \qquad (2)$$

The 95%-CIs of the estimated parameters are computed from the curvature of the log-likelihood function at the maximum. When replicated data is available, we multiply the likelihood function across the replicates. The replicates share the same parameters $\alpha, \beta, \rho_0^i, a^i$. Furthermore, when there are only two time points, ACIDES has an option to set $\alpha$ and $\beta$ as a function of time. This option was used to generate the results of Fig. 5.

## Synthetic data
Synthetic count data $n_t^i$ ($i = 1, 2, ..., M$, $t \in T$) are generated from the model (2) for a given parameter set $\alpha, \beta, \rho_0^i, a^i$ ($i = 1, 2, ..., M$). For Fig. 1, we use $\alpha, \beta = 0.69, 0.8$ with $(a^i, \log \rho_0^i)$ generated from the normal distribution with the expected values $(-1, 1)$ and the standard deviations $(0.25, 1)$. $(M, N_t)$ are $(5 \times 10^4, 10^7)$ for the data-rich case (Fig. 1e) and $(10^6, 10^6)$ for the data-poor one (Fig. 1f).

## Model inference
To maximize the likelihood function, we develop a two-step algorithm. The first step infers $(\rho_0^i, a^i)$, while the second $(\alpha, \beta)$ and then we iterate the two steps until convergence is reached. All inferences are done with a gradient descent algorithm, and to reach convergence 10-30 iterations are usually sufficient. The first step is itself iterative, and loops between the inference of $(\rho_0^i, a^i)$ and $C_t$ by treating $C_t$ as a parameter. Here we also introduce a gauge choice because of the redundancy between $\rho_0^i$, $a^i$ and $C_t$ (the caption of Fig. S2 for more details). In the second step, the inference of $(\alpha, \beta)$ with a straightforward gradient method produces a bias (Fig. S2E). In order to correct this, at each iteration the algorithm adopts a teacher-student framework, runs a simulation of the count data with the current parameters to obtain an estimation of the bias, which is then used to correct the real inference and update the parameters.

In order to reduce computational time and to increase the stability of the algorithm, we first run the inference algorithm on a subset of variants to estimate $\alpha$, $\beta$. We then compute $(\rho_0^i, a^i)$ of the excluded variants using the estimated $\alpha$, $\beta$. For this subset, we use the variants that satisfy the following two criterions: (i) their counts are larger than 0 more than twice in the selection rounds and (ii) whose total NGS count (as summed over all the rounds) is above a threshold. We set this threshold to 100 for all the datasets except for Data-E -G, where 10000 is used. This is because the noise in these experiments is larger than the others. Results are stable by changing the threshold value (Fig. S2F).

## Simulated rank and rank robustness (RR)
Using the standard deviations $\delta a^i$ ($i = 1, ..., M$) of estimated scores $a^i$, we discard the variants with higher estimated errors. We keep 5000 variants for further analysis and denote by $A$ their indices. We then rearrange the variant index in $A$ in descending order of $a^i$ to define a naive rank (the $x$-axis of Fig 2a–g). To obtain a corrected rank (the $y$-axis of Fig 2a–g), we first generate synthetic scores using the normal distribution with the expected value $(a^i)_{i \in A}$ and the standard deviation $(\delta a^i)_{i \in A}$. Based on the generated scores, we rearrange the variant index in descending order and define a synthetic naive rank. Repeating this estimation 3000 times, we then compute the median and 95%-CI of the obtained synthetic naive ranks. This 95%-CI is defined as the corrected rank.

To estimate RR, we compare the top-50 variants in the naive rank and each synthetic naive rank. We count the number of overlaps between them and average it over the 3000 estimations. RR is computed by dividing the obtained overlap by 50.

## NGS-Downsampling for Fig. 3
To obtain downsampled count data $\tilde{n}_t^i$ ($i = 1, 2, ..., M$, $t \in T$) by a factor $\epsilon$, we sample synthetic data from the likelihood function (2) with a reduced number of the total counts $\epsilon N_t$ ($t \in T$) and with the estimated parameters $\rho_0^i$, $a^i$, $\alpha$, $\beta$ ($i = 1, 2, ..., M$). To obtain a downsampled RR in Fig. 3, we first re-estimate $a^i$ ($i = 1, 2, ..., M$) from $\tilde{n}_t^i$ ($i = 1, 2, ..., M$, $t \in T$) using the values of $(\alpha, \beta)$ that are already known, and then perform the estimation of RR described above. Using the synthetic data, we show that this downsampling method captures well the RR of actual NGS-read-reduced data (Fig. S5). Practically for Fig. 3, we rescale the downsampled RR using the RR obtained in a similar manner, but with $\epsilon = 1$. This is to remove a minor bias due to re-sampling and to focus on the trend in NGS reads.

## Cross validation test for Fig. 5
For Fig. 5, we perform a leave one out cross-validation test[48]. Below we assume that there are only two time points in the data, which is denoted by $t = 0$ (input) and $t = 1$ (output). We first separate the replicated data into one test data and the remaining training data. To the training data, we apply ACIDES and infer the model parameters $\alpha, \beta, \rho_0^i, a_0^i$. Based on these parameters, using the likelihood function (2), we re-sample the count data $n_t^i$ and compute the mean and standard deviation of an enrichment $f^i$ for every $i$. Here the enrichment $f^i$ is defined as

$$f^i = \log\left(\frac{n_1^i}{n_0^i} \times \frac{n_{\text{wt}}^0}{n_{\text{wt}}^1}\right), \qquad (3)$$

where $n_{\text{wt}}^i$ is the count data for the wild type variant. Similarly, for the test data, we apply ACIDES with $\alpha, \beta$ obtained in the training data and estimate $\rho_0^i, a_0^i$. These parameters are then used to re-sample the count data $n_t^i$ and compute the mean and standard deviation of an enrichment $f^i$ for every $i$. For the other algorithms, such as DiMSum[48], Enrich2[41], the count-based model, the Bayesian regularization model[74], and standard-deviation based model, we follow[48] to compute the enrichments. While ACIDES uses resampling of count data based on the negative-binomial error model to estimate the enrichment statistics, the others rely on either a Poisson-based approximation (count-based model), empirical variance of fitness (Bayesian regularization model, standard-deviation based model), or both (DiMSum, Enrich2).

The $z$-score is then defined as

$$z^i = \frac{\bar{f}_{\text{training}}^i - \bar{f}_{\text{test}}^i}{\sqrt{(\sigma_{\text{training}}^i)^2 + (\sigma_{\text{test}}^i)^2}}, \qquad (4)$$

where $\bar{f}_{\text{training}}^i, \bar{f}_{\text{test}}^i$ are the mean enrichments obtained from the training data and test data, while $\sigma_{\text{training}}^i, \sigma_{\text{test}}^i$ are the standard deviation of enrichment obtained from the training data and test data.

The statistics of $z^i$ over all the variants $i$ is investigated using the following two quantities; (i) the inverse of the standard deviation of $z^i$ from 1, as indicated by the red box plots of Fig. 5 (more precisely, 1 minus the deviation of this quantity from 1), and (ii) the coefficient of determination $R^2$ obtained from quantile-quantile plot, as indicated by the blue box plots of Fig. 5. For (ii), we first plot a quantile-quantile plot (c.f., Fig. 3a of ref. 48) where $x$-axis is the quantiles obtained from the standard normal distribution and $y$-axis is the estimated quantiles. The closeness of the line to $y = x$ is then measured using the coefficient of determination $R^2$ between $y = x$ line and this plot.

## Pre-processing of Data-E and Data-G
In their original datasets, Data-E and Data-G contain a large number of variants whose total counts are very low (but not zero). In order to speed up the analysis and make the analysis more robust we removed the variants whose total counts are smaller than 1000 (Data-E) and

than 100 (Data-G). The NGS depth and the number of unique variants shown in Table 1 are after this preprocessing.

## Statistics & Reproducibility

In this study, we use maximum likelihood estimations to compute the scores of each variant for each of publicly available dataset. The statistical errors of the scores are estimated based on the curvature of the log likelihood function at the maximum. When estimating the corrected rank and its 95%-CI, we use 3000 bootstrap samples. By varying this value, we confirmed that the sample sizes do not affect the outcome.

## Input datafile

The input for ACIDES is a count dataset matrix. Each row corresponds to an index (e.g., barcode) for each variant, while each column indicates the experimental round of DE or DMS. The entries of the matrix represent the count data observed in NGS for each variant in each round. This can be generated from the FASTQ file, following preprocessing depending on the experimental details. In general, several steps may be required, such as trimming constant regions, removing erroneous sequences that do not contain these regions, and conducting quality control based on the Q-score. These procedures can be done using tools such as Cutadapt. In the case of paired-end reads, it is also necessary to merge these reads beforehand, which can be achieved with tools like PEAR. Overall, the DiMSum pipe line offers all these procedures in one package.

## Reporting summary

Further information on research design is available in the Nature Portfolio Reporting Summary linked to this article.

## Data availability

The data listed in Table 1 are publicly available. We have also provided the corresponding count data matrices for Data-A through Data-G in our Github repository [https://github.com/nemoto-lab/ACIDES/tree/main/data], which can be used directly as input files for ACIDES. The random-peptide inserted library used for Fig. S1 can be downloaded from SRR26390523 [https://www.ncbi.nlm.nih.gov/sra/SRR26390523]. Source data are provided with this paper.

## Code availability

ACIDES is available on Github: https://github.com/nemoto-lab/ACIDES/ under the GNU GPLv3.0 license. It comes with a tutorial that outlines how to use ACIDES along with a working Python code. The default parameters of ACIDES are used for the analysis in this article, unless otherwise specified. The version of ACIDES used in this article, ACIDES-v0, is archived on Zenodo[75].

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

## Acknowledgements

The authors would like to thank A. Rubin and D. Fowler for kindly providing them with datasets (Data-A and Data-B) and a working code for Enrich2. The authors also would like to thank L. C. Byrne and T. Mora for

useful comments and discussions, M. Desrosiers and C. Robert for their technical assistance with the production of plasmids and viral vectors, and O. Marre for facilitating and initiating the collaboration and helpful discussions. This work was supported by ERC Starting Grant (REGE-NETHER 639888 to D.D.), European Research Council (ERC) Horizon 2020 Framework Programme Project (863214 – NEUROPA to D.D.), UNADEV, the Institut National de la Santé et de la Recherche Médicale (INSERM), Sorbonne Université (to D.D. and U.F.), The Foundation Fighting Blindness, Agence National de Recherche (ANR) RHU Light4-Deaf, LabEx LIFESENSES (ANR-10-LABX-65 to D.D.), IHU FOReSIGHT (ANR-18-IAHU-01 to D.D.), JSPS KAKENHI (Grant Number 22K17994 to T.N.), World Premier International Research Center Initiative (WPI), MEXT, Japan (to T.N.), and Paris Region Postdoctoral Fellowship (PRPF to E.Z.).

## Author contributions

U.F., D.D., and T.N. designed research, obtained funding. T.N. and T.O. performed numerical research. A.P., M.T. and E.Z. performed experimental research. All authors analyzed data, contributed to discussions. T.N. and U.F. prepared the initial draft of the manuscript, and all of the authors reviewed it.

## Competing interests

The authors declare no competing interests.
