## [Peer Review File · Nature Communications]

ACIDES: on-line monitoring of forward genetic screens for protein engineeringReviewer #1 (Remarks to the Author):

The authors present a new software package, ACIDES, for analyzing directed evolution (DE) and deep mutational scanning (DMS) data. The paper demonstrates several improvements over previous work in the field, specifically Enrich2, and presents an innovative and novel approach to analyzing datasets with multiple time points. However, the manuscript needs some updates to address recent developments in the field.

Major Comments

1. Enrich2 is no longer the state-of-the-art method. ACIDES uses the negative binomial, which underpins the analysis of differential gene expression, but they are not the first to do so. The authors should compare their results with DiMSum (PMID 32799905) and discuss the similarities and differences between the two methods in detail.
2. If the ACIDES_module.py file included in the download is the complete implementation of ACIDES, it's important to point out to readers (as your potential users!) that this requires another piece of software (such as Enrich2) to generate the count files to be analyzed. It's fine that ACIDES is not a full pipeline, but the authors should provide instructions in the paper for how a new dataset can be analyzed using a combination of ACIDES and other programs.
3. All of the datasets analyzed in the paper are quite old. I wonder if the advantages of ACIDES in dealing with sampling error and other sources of noise would generalize to "better behaved" datasets? That is, does ACIDES perform well because the variant libraries they analyzed were highly non-uniform in their starting proportions? This could be answered by analyzing some newer datasets, or by simulation of datasets with different amounts of various types of noise, or both.

Minor Comments

1. In the Figure 1 legend, the authors state that "The higher the score, the better the variant for the task" but this is not always the case. The relationship between population-level enrichment and variant function depends on the assay. For example, a published dataset testing beta-amyloid variants for their ability to aggregate would be oriented in the opposite way, since aggregation is lethal to cells. The authors should be careful about this kind of phrasing and check that more generic phrasing is used throughout the manuscript.
2. The authors say that ACIDES is suitable for DE and certain DMS experiments, specifically phage-display and yeast two-hybrid. Why is ACIDES not suitable for cell-based DMS experiments? These are much more common, and there are numerous examples in the literature using yeast, human, and bacterial cells.
3. The authors say that they'll put their code on GitHub when the paper is ready for publication. Please also use an archival service like Zenodo that will provide a DOI for each software release.
4. In the first paragraph of the results the authors cite major differential expression papers that use the negative binomial, but it would be helpful to readers familiar with bioinformatics to state that this is a method that's been popularized for that type of analysis.
5. The introduction would benefit from the addition of some more recent citations, as most of the DE and DMS studies cited are quite old.

Reviewer #2 (Remarks to the Author):

In this study, the authors develop an advanced method for determining the fitness rank of individual variants associated with directed evolution or deep mutational scanning studies. Their statistical approach to estimating the next-gen sequencing (NGS) sampling noise uses a negative binomial distribution with parameters inferred via maximum likelihood. Being able to accurately rank the fitness of variants within a fitness-matured population (e.g. iterative rounds of selection)

is of high importance, especially with the rise of data-hungry machine learning algorithms. The authors use publicly available datasets as well as a newly generated random-peptide dataset to compare their methodology against the reputable Enrich2 algorithm. The performance of ACIDES and Enrich2 is compared by using each approach to analyze the ranking of variants taken from multiple replicates (Figure 4).

The authors indicate that the ground-truth is known for some data, but it is not well described how the ground truth rankings were obtained. Were individual variants characterized via wet-lab experiments? It may not be sufficient to use selectivity consistency as a metric for accuracy if the actual (i.e. experimentally validated) ranks are yet to be measured.

It would be informative to the reader if the authors included in the supplement a table of the calculated alpha and beta values (negative binomial parameters) associated with each dataset.

Regarding the ACIDES_module.py script itself, the authors are encouraged to add brief descriptions for each defined function to assist with interpretability.

Reviewer #3 (Remarks to the Author):

In this manuscript, Nemoto et al. developed ACIDES, a tool to assist directed evolution campaigns that apply NGS analyses for monitoring variant enrichment during multiple rounds of selection. One key improvement over the SOTA methods, such as Enrich2, is the use of log-likelihood maximization and a softmax function for score inferring, as well as a negative binomial distribution to model variability. The authors applied the resulting statistical errors to adjust the naïve rank, computing the reliability of variant ranking as Rank Robustness (RR). The authors demonstrated that RR may serve as a good indication of "selection convergence" for informed decisions on additional experiments such as further selection rounds, deeper NGS coverages, and more replicates. This manuscript was well organized, and the results are clearly presented.

The primary concern is that RR only evaluates the changes in variant ranking of selectivity before and after considering statistical errors. However, one ultimate goal of data analysis on selection-seq is to estimate the fitness of an individual variant by calculating its selectivity/enrichment in an evolving population under selection pressure. It is important to assess how well ACIDES results are consistent with the "ground truths"—the fitness ranking of variants in a second-pass confirmation after the pooled selection. Do the corrected rank produced by ACIDES show better correlation with the ground truths than the naïve rank, or another rank calculated by other methods, such as Enrich2? Will RR be a good indication in this case?

Overall, this manuscript presents a useful data tool for the NGS-coupled selection strategy in protein engineering. In addition to the primary concern, I also have a few technical comments.

1. Directed evolution (DE) often involves iterations of both genetic mutagenesis and phenotypic selection/screening. However, in several instances, the authors use the term DE to refer to one round of genetic mutagenesis followed by a single series of continuing rounds of selection/enrichment. It is unclear whether ACIDES can deal with newly introduced genetic mutations that do not exist in the original library. I suggest that the authors carefully examine every instance of DE to ensure the correct use of concept.
2. The analyses on the high, intermediate, and low RR data sets were superficial, and the data presentation was sometimes confusing.
 - 2.1. Figure 2 and Table 1: the current order makes it difficult to discern trends. I recommend reordering the data sets in a descending sequence of RR.
 - 2.2. The authors emphasized the importance of sufficient NGS depths relative to the variant library sizes. Therefore, I suggest they include a column of Reads/# variants in Table 1.
 - 2.3. While NGS depths relative to library sizes appear to predict the RR values of the synthetic libraries in this study (Fig. 1E-G), as well as the high AB and intermediate FG groups (Fig. 2), the low CD group did not follow this trend at all. The authors simply attributed this discrepancy to the intrinsic difficulty of these in-vivo experiments, which is not entirely convincing. For example, it is possible that the introduction of new mutations by error-prone PCR during DE is responsible for

the poor performance of ACIDES on Data C. Therefore, I recommend that the authors conduct a more in-depth analyses of Data-C and -D to understand the reasons for the low RRs. For example, a similar analysis to that in Figure 3 on Data-C and -D could help to determine whether further selection rounds or deeper NGS coverage would improve the results.

3. It would be helpful for the authors to discuss the potential applicability of ACIDES to other types of NGS-based DMS data sets, particularly those generated by FACS-seq. Additionally, it is not clear why only a small number of data sets were chosen for benchmarking, given the numerous reported NGS-based DMS data sets available. Please explain why the select set is of sufficient representation.

4. Benchmarking with Enrich2

4.1. It is not clear why only Data-A and Data-B were utilized for comparing ACIDES and Enrich2, while the performance of ACIDES on Data sets C-G was much worse. The authors implied that the experiments of Data sets C-G were either not well executed or intrinsically more difficult, but it is possible that ACIDES is only effective for certain types of tasks that are similar to those performed for Data-A and -B?

4.2. For method evaluation, the authors utilized only the scoring consistency among replicates, but there are more evaluation metrics. For example, it would be valuable to examine how ACIDES and Enrich2 scores and errors correlate with each other, particularly for Data C-G. Additionally, if secondary confirmation data are available in the literature, it would be worthwhile to investigate how ACIDES and Enrich2 ranks correlate with ground truths.

4.3. Enrich 2 can estimate variant ranks from only one enrichment round (two timepoints). For this type of data, will ACIDES outperform Enrich2 as well?

Authors' response and summary of changes

Reviewer #1 (Remarks to the Author):

The authors present a new software package, ACIDES, for analyzing directed evolution (DE) and deep mutational scanning (DMS) data. The paper demonstrates several improvements over previous work in the field, specifically Enrich2, and presents an innovative and novel approach to analyzing datasets with multiple time points. However, the manuscript needs some updates to address recent developments in the field.

We thank the reviewer for their positive comments, careful review of our manuscript, and constructive suggestions. We will address each comment/question point by point below. Please note that the referees' comments are written in ***italic black***, our answers in **blue**, and any text referring to modifications of the manuscript in **red**. Corresponding changes in the manuscript are also highlighted in **red**.

Major Comments

1. Enrich2 is no longer the state-of-the-art method. ACIDES uses the negative binomial, which underpins the analysis of differential gene expression, but they are not the first to do so. The authors should compare their results with DiMSum (PMID 32799905) and discuss the similarities and differences between the two methods in detail.

We would like to thank the reviewer for pointing us towards DiMSum. As we previously focused only on datasets with multiple time-points; we highly appreciate this suggestion. In the revised version of our manuscript, we have generalized ACIDES to handle datasets with two time points and multiple replicates, and we have incorporated a new figure (Fig.5) to compare ACIDES with DiMSum [Faure et al. 2020]. For this comparison, we used 12 new datasets that were originally used in the DiMSum paper and employed the same cross validation technique as explained as follows. Each dataset has at least 3 replicates. Following [Faure et al. 2020], we performed leave-one-out cross-validation by using one replicate for test and others for training. The quantity to compare is the statistics of the z-score of enrichments. (Please refer to the Methods section of the manuscript for more details. The key idea here is that the closer the statistics are to the standard normal distribution, the better the method is.) In the following figure, we use the red box plot to compare the inverse values of the standard deviation for the z-score, the same metrics used in the leave-one-out cross-validation of DimSum paper [Fig. 3e in Faure et al. 2020].

(Note that, in this figure, the closer the value is to 1, the better the method's performance.) As such, ACIDES and DiMSum have scores close to 1, with superior performance with respect to Enrich2. DiMSum uses a similar idea to ACIDES. It models large dispersion of enrichment noise by modifying the Poisson error. The advantage of ACIDES over DiMSum is the following:

1. DiMSum can only be applied to two-point time data with replicates, while ACIDES can also be applied to multiple-point time data with or without replicates.
2. DiMSum modifies the Poisson error by multiplying a factor, and this factor is determined to optimize the prediction of the standard deviation of enrichments. On the other hand, ACIDES uses the negative binomial distribution, which is a natural extension of the Poisson distribution to take into account overdispersion. The fitting is also done by maximizing the log-likelihood function. This difference appears, for example, when we look at the statistics of z-score beyond the standard deviation. In the figure above, we also plot the coefficient of determination (R² score) as the blue box plot, defined using the quantile-quantile plot for the z-score of the enrichment. (The quantile-quantile plot (c.f., Fig.3a in Faure et al. 2020) compares the shape of the z-score probability distribution with that of the standard normal distribution, by plotting the theoretical quantiles from the normal distribution in the x-axis and the observed z-score quantiles in the y-axis. In this context, R² score quantifies how well the points in the plot align with the y=x line. Consequently, this R² score reflects the extent of similarity to the normal distribution, going beyond just comparing standard deviations.) The blue box plot shows that ACIDES slightly outperforms DiMSum and remains better than the other previous methods. (Again, in this figure, 1 is the maximum performance.)

We have included this new figure (Fig. 5) in the manuscript and updated the corresponding section, “Comparison with previous work”, as well as the tutorial file to explain this comparison between ACIDES and the other algorithms.

2. If the `ACIDES_module.py` file included in the download is the complete implementation of ACIDES, it's important to point out to readers (as your potential users!) that this requires another piece of software (such as Enrich2) to generate the count files to be analyzed. It's fine that ACIDES is not a full pipeline, but the authors should provide instructions in the paper for how a new dataset can be analyzed using a combination of ACIDES and other programs.

We agree that it is crucial to guide readers on generating a file with variant counts from a FASTQ file. We now have added a paragraph in the “Code availability” section detailing how to perform these analyses, citing Cutadapt, PEAR, and DIMSum as references.

3. All of the datasets analyzed in the paper are quite old. I wonder if the advantages of ACIDES in dealing with sampling error and other sources of noise would generalize to “better behaved” datasets? That is, does ACIDES perform well because the variant libraries they analyzed were highly non-uniform in their starting proportions? This could be answered by analyzing some newer datasets, or by simulation of datasets with different amounts of various types of noise, or both.

Following this advice, we've analyzed 12 additional datasets, 11 of which were published after 2017. Using these datasets, we compared the performance of ACIDES with five other algorithms, including DiMSum and Enrich2. We have added these new analyses in Fig. 5 of the main text.

The question about “better behaved” datasets is certainly interesting. We thus attempted to study the correlation between the performance of the algorithms (ACIDES, DiMSum, Enrich2, etc...) and the standard deviation of counts in the initial library by using all these new datasets, but we didn't find any significant correlation (see Figure below).

Minor Comments

1. In the Figure 1 legend, the authors state that “The higher the score, the better the variant for the task” but this is not always the case. The relationship between population-level enrichment and variant function depends on the assay. For example, a published dataset testing beta-amyloid variants for their ability to aggregate would be oriented in the opposite way, since aggregation is lethal to cells. The authors should be careful about this kind of phrasing and check that more generic phrasing is used throughout the manuscript.

We agree with the reviewer on this matter. A high score for a variant implies that it outperforms others in executing the selected task. However, it doesn't automatically mean that the mutation is beneficial, as the beta-amyloid example demonstrates. We have rephrased the corresponding sentences in the main text (in the caption of Fig.1 and in a paragraph before Section “Integration into the experimental pipeline”) accordingly.

2. The authors say that ACIDES is suitable for DE and certain DMS experiments, specifically phage-display and yeast two-hybrid. Why is ACIDES not suitable for cell-based DMS experiments? These are much more common, and there are numerous examples in the literature using yeast, human, and bacterial cells.

The reviewer is right, they were intended as examples, but the text was indeed misleading. We thank the reviewer for this useful comment to improve our manuscript. We have rephrased this sentence in the last paragraph of Introduction and added references for the cell-based DMS.

3. The authors say that they'll put their code on GitHub when the paper is ready for publication. Please also use an archival service like Zenodo that will provide a DOI for each software release.

We will also use Zenodo. We thank the reviewer for this suggestion.

4. In the first paragraph of the results the authors cite major differential expression papers that use the negative binomial, but it would be helpful to readers familiar with bioinformatics to state that this is a method that's been popularized for that type of analysis.

We appreciate the suggestion. We have added this explanation to the first paragraph of Results. .

5. The introduction would benefit from the addition of some more recent citations, as most of the DE and DMS studies cited are quite old.

We thank the reviewer for this suggestion, we have cited more recent articles in the first paragraph of the introduction:

- C Anders Olson, Nicholas C Wu, and Ren Sun. A comprehensive biophysical description of pairwise epistasis throughout an entire protein domain. Current biology 24, 2643 (2014).

- Guillaume Diss and Ben Lehner. The genetic landscape of a physical interaction. *Elife* 7, e32472 (2018).
- Benedetta Bolognesi, Andre J Faure, Mireia Seuma, Jo ðrn M Schmiedel, Gian Gaetano Tartaglia, and Ben Lehner. The mutational landscape of a prion-like domain. *Nature communications* 10, 4162 (2019).
- Andre J Faure, Ju ´lia Domingo, J ðrn M Schmiedel, Cristina Hidalgo-Carcedo, Guillaume Diss, and Ben Lehner. Mapping the energetic and allosteric landscapes of protein binding domains. *Nature* 604, 175 (2022).
- Ju ´lia Domingo, Guillaume Diss, and Ben Lehner. Pair- wise and higher-order genetic interactions during the evolution of a tRNA. *Nature* 558, 117 (2018).
- Chuan Li and Jianzhi Zhang. Multi-environment fitness landscapes of a tRNA gene. *Nature ecology & evolution* 2, 1025 (2018).
- Kyryn R Hanning, Mason Minot, Annmaree K Warren- der, William Kelton, and Sai T Reddy. Deep mutational scanning for therapeutic antibody engineering. *Trends in pharmacological sciences* 43, 123 (2022).
- Tyler N Starr, Allison J Greaney, Sarah K Hilton, Daniel Ellis, Katharine HD Crawford, Adam S Dingens, Mary Jane Navarro, John E Bowen, M Alejandra Tor- torici, Alexandra C Walls, et al. Deep mutational scanning of SARS-CoV-2 receptor binding domain reveals constraints on folding and ACE2 binding. *cell* 182, 1295 (2020).
- Tyler N Starr, Allison J Greaney, William W Hannon, Andrea N Loes, Kevin Hauser, Josh R Dillen, Elena Ferri, Ariana Ghez Farrell, Bernadeta Dadonaite, Matthew Mc- Callum, et al. Shifting mutational constraints in the SARS-CoV-2 receptor-binding domain during viral evolution. *Science* 377, 420 (2022).
- Liselot Dewachter, Aaron N Brooks, Katherine Noon, Charlotte Cialek, Alia Clark-ElSayed, Thomas Schalck, Nandini Krishnamurthy, Wim Vers ´ees, Wim Vranken, and Jan Michiels. Deep mutational scanning of essential bacterial proteins can guide antibiotic development. *nature communications* 14, 241 (2023).

Reviewer #2 (Remarks to the Author):

In this study, the authors develop an advanced method for determining the fitness rank of individual variants associated with directed evolution or deep mutational scanning studies. Their statistical approach to estimating the next-gen sequencing (NGS) sampling noise uses a negative binomial distribution with parameters inferred via maximum likelihood. Being able to accurately rank the fitness of variants within a fitness-matured population (e.g. iterative rounds of selection) is of high importance, especially with the rise of data-hungry machine learning algorithms. The authors use publicly available datasets as well as a newly generated random-peptide dataset to compare their methodology against the reputable Enrich2 algorithm. The performance of ACIDES and Enrich2 is compared by using each approach to analyze the ranking of variants taken from multiple replicates (Figure 4).

We thank the reviewer for their positive comments, careful review of our manuscript and constructive suggestions. We will address each comment/question point by point below.

Please note that the referees' comments are written in **italic black**, our answers in **blue**, and any text referring to modifications of the manuscript in **red**. Corresponding changes in the manuscript are also highlighted in **red**.

The authors indicate that the ground-truth is known for some data, but it is not well described how the ground truth rankings were obtained. Were individual variants characterized via wet-lab experiments? It may not be sufficient to use selectivity consistency as a metric for accuracy if the actual (i.e. experimentally validated) ranks are yet to be measured.

In Fig. 1E & F we used synthetic data, which was generated based on a negative binomial distribution. As a result, we knew the ground-truth value a priori. This served as a test to evaluate the performance of our inference framework under idealized situations. We acknowledge that the initial manuscript did not sufficiently clarify this point, so we have added more explanations in the second paragraph of the result section. In addition to this, using the synthetic data, we systematically compare ACIDES with Enrich2 by measuring the coefficient of determination (R²) between the estimated rank and the ground truth. We confirm the advantage of ACIDES over a range of RR with various scenarios:

In this figure, N is the total NGS reads and M is the number of sequences. . This result is now displayed in the Supplementary Information (Fig.S8).

As for the usage of selectivity consistency in Fig.4, the aim of this test is to assess the accuracy of the (model-based) estimated errors associated with the scores. For this purpose, we believe that the use of selectivity consistency fulfills its purpose: Consistency should be high for sequences with small predicted errors if these predictions are indeed accurate. We acknowledge that the original text did not sufficiently clarify this point and may have been misleading. We have modified the first paragraph in the “Comparison with previous work” section accordingly.

It would be informative to the reader if the authors included in the supplement a table of the calculated alpha and beta values (negative binomial parameters) associated with each dataset.

We have added their values to the Supplementary Table S1:

Label	A	B	C	D	E	F	G
alpha	0.59	0.87	0.15	0.01	0.15	0.77	0.03
beta	0.97	0.24	5.10	23.12	0.19	0.28	0.16

Regarding the ACIDES_module.py script itself, the authors are encouraged to add brief descriptions for each defined function to assist with interpretability.

We apologize for not including many comments to explain this test code. In the re-submitted manuscript we now include an updated version with more comments and descriptions, which can be downloaded from this link:

<https://www.dropbox.com/sh/kn4d5443y980n4b/AADZZIxQtGeHnmzOAHMc7Sa2a?dl=0>

Reviewer #3 (Remarks to the Author):

In this manuscript, Nemoto et al. developed ACIDES, a tool to assist directed evolution campaigns that apply NGS analyses for monitoring variant enrichment during multiple rounds of selection. One key improvement over the SOTA methods, such as Enrich2, is the use of log-likelihood maximization and a softmax function for score inferring, as well as a negative binomial distribution to model variability. The authors applied the resulting statistical errors to adjust the naïve rank, computing the reliability of variant ranking as Rank Robustness (RR). The authors demonstrated that RR may serve as a good indication of “selection convergence” for informed decisions on additional experiments such as further selection rounds, deeper NGS coverages, and more replicates. This manuscript was well organized, and the results are clearly presented.

We thank the reviewer for their positive comments, careful review of our manuscript and constructive suggestions. We will address each comment point by point below. Please note that the reviewer’s comments are written in **italic black**, our answers in **blue**, and any text referring to modifications of the manuscript in **red**. Corresponding changes in the manuscript are also highlighted in **red**.

Please also note that, as requested in point 2.1, we have reordered the 7 datasets in Fig 2 & Table 1 in the new version of the manuscript, arranging them in descending order of Rank Robustness (RR). However, in order to enhance the readability of the exchanges between

the reviewer comments and our replies, we keep using the old labels in this response. A conversion table is provided below that might be helpful:

Old label (in this reply)	A	B	C	D	E	F	G
New label (in the manuscript)	A	B	E	G	F	D	C

The primary concern is that RR only evaluates the changes in variant ranking of selectivity before and after considering statistical errors. However, one ultimate goal of data analysis on selection-seq is to estimate the fitness of an individual variant by calculating its selectivity/enrichment in an evolving population under selection pressure. It is important to assess how well ACIDES results are consistent with the “ground truths”—the fitness ranking of variants in a second-pass confirmation after the pooled selection. Do the corrected rank produced by ACIDES show better correlation with the ground truths than the naive rank, or another rank calculated by other methods, such as Enrich2? Will RR be a good indication in this case?

We thank the referee for raising these key questions. To answer them, we demonstrate that the corrected rank shows a stronger correlation with the true rank compared to both naive rank and Enrich2 rank. To do so, we systematically analyzed synthetic datasets, where the true rank was known a priori. For various scenarios (high/low total NGS reads with high/low total number of sequences), we estimated the R2 score to measure the correlation between the true rank and the ranks estimated by different methods: Enrich2 (red), ACIDES naive rank (blue), and ACIDES corrected rank (green). These results are plotted in the following figure (N is the total NGS reads and M is the number of unique sequences):

As can be seen in this figure, for a range of RR, ACIDES rank shows better R2 over Enrich2 rank and naive rank.

RR is a measure of the deviation between the true rank and the naive rank. In the figure above, we can see that RR anticorrelates with the gain of ACIDES over Enrich2: the smaller the RR is, the more advantage ACIDES has over Enrich2. **We've added this analysis to compare ACIDES and Enrich2 to the supplementary information (Fig.S8).**

In addition to this demonstration, below we add an explanation of why the corrected rank performs better than the naive one by using a simple example. Suppose we identify our top 10 variants based on naive rank, yet the differences in estimated selectivities among these variants are smaller than their estimated statistical errors (please refer to the following figure). In such a situation, the true ranks of these variants are in average higher than their naive ranks:

This is because other variants, not included in the naive top 10, could be the real top performers. The corrected rank addresses this discrepancy by taking into account the statistical error using bootstrapping analysis. In this procedure, we resample the scores using the estimated selectivity and its standard deviation based on the normal distribution, and re-estimate the naive rank:

Average of resampled rank = corrected rank
 > naive rank (for top performing variants)

These re-estimated ranks are generally higher (or lower) than the naive rank for the top (or bottom) half of the variants. Therefore, if we focus on the top variants identified in the naive rank, their corrected rank correlates better with the true rank.

Overall, this manuscript presents a useful data tool for the NGS-coupled selection strategy in protein engineering. In addition to the primary concern, I also have a few technical comments.

1. Directed evolution (DE) often involves iterations of both genetic mutagenesis and phenotypic selection/screening. However, in several instances, the authors use the term DE to refer to one round of genetic mutagenesis followed by a single series of continuing rounds of selection/enrichment. It is unclear whether ACIDES can deal with newly introduced genetic mutations that do not exist in the original library. I suggest that the authors carefully examine every instance of DE to ensure the correct use of concept.

The referee is indeed right, in its current version ACIDES can not properly account for additional rounds of mutagenesis (for example, by error-prone pcr) between the selection rounds. Similar to Enrich2, when this happens, ACIDES will treat the newly generated sequences as if they had zero counts at the beginning, and the scores for these newly generated variants may be slightly underestimated. Note that, as the error rate of error-prone PCR is less than a few percent per position, in most cases, this has little practical impact on variants that already existed. Consequently, the only concern is that we need to be cautious about any interesting variants that begin to emerge after the error-prone PCR.

In our data, only Data-C uses an error-prone PCR following the third round of selections. However, we confirmed that all the top-50 variants observed in Data-C were already present from the start of the experiment. **We've added these explanations to the discussion section (in the second to last paragraph).**

2. The analyses on the high, intermediate, and low RR data sets were superficial, and the data presentation was sometimes confusing.

The referee is right, our analysis of high, intermediate and low RR was superficial and confusing. We have now modified the text, rearranged the dataset and added an in-depth analysis of noise and score distribution in the experiments (please refer to the following points). We hope that the referee finds the updated section clearer and more insightful.

2.1. Figure 2 and Table 1: the current order makes it difficult to discern trends. I recommend reordering the data sets in a descending sequence of RR.

We have now reordered the datasets in a descending sequence of RR.

2.2. The authors emphasized the importance of sufficient NGS depths relative to the variant library sizes. Therefore, I suggest they include a column of Reads/# variants in Table 1.

We have now added avg. count in Table 1 which measures the mean reads per variant per round.

2.3. While NGS depths relative to library sizes appear to predict the RR values of the synthetic libraries in this study (Fig. 1E-G), as well as the high AB and intermediate FG groups (Fig. 2), the low CD group did not follow this trend at all. The authors simply attributed this discrepancy to the intrinsic difficulty of these in-vivo experiments, which is not entirely convincing. For example, it is possible that the introduction of new mutations by error-prone PCR during DE is responsible for the poor performance of ACIDES on Data C. Therefore, I recommend that the authors conduct a more in-depth analyses of Data-C and -D to understand the reasons for the low RRs. For example, a similar analysis to that in Figure 3 on Data-C and -D could help to determine whether further selection rounds or deeper NGS coverage would improve the results.

We appreciate this inquiry. The low RR in the CD group is indeed an important issue that warrants further discussion. First, we now believe that the error-prone PCR is not the reason for these small RR. This is because (i) data D and E do not include error-prone PCR, yet they still show low RR, and (ii) in data C, which includes error-prone PCR after the third round of selections, the top 50 variants identified by ACIDES were present from the beginning, thus being less affected by these additional mutagenesis. (Please also refer to the reply to comment 1 for more details on the impact of error-prone PCR.)

To better understand why Data-C, D and E show low RR, and to gain a more comprehensive understanding of RR in general, we have computed two additional insightful metrics: (i) Accuracy: This measures the accuracy of the score measurements, calculated based on the inverse of average errors predicted by the model for the estimated scores. And (ii) Spread: This measures the extent to which the variants are intrinsically different in their performance for the task, computed as the normalized difference between the estimated scores of the highest-scoring variant and the variant ranked 1000th. The larger these quantities are, the easier it becomes to distinguish the best-performing variants, thus resulting in a higher RR:

	Data A	Data B	Data C	Data D	Data E	Data F	Data G
Accuracy	3.06	1.83	0.44	0.34	1.22	1.40	0.95
Spread	0.83	0.74	1.88	2.00	0.54	0.77	0.96
RR	0.88	0.82	0.55	0.47	0.54	0.71	0.74

It is interesting to find out that the reasons for the low RR values for Data-C, Data-D and Data-E are different. Data-C and Data-D (DE experiments using AAV) suffer from very low Accuracies in their experiments, even though they have large Spreads. On the other hand, Data-E has a relatively high Accuracy, but Spread is the smallest, meaning that their variants perform similarly in their experiment. These two factors counterbalance each other, resulting in similar RR values for these datasets. By examining these two quantities, we can also understand the behavior of Data A, Data-B, Data F, and Data-G. They all have similar Spreads, which may be because they follow similar experimental protocols. The Accuracy of Data F and Data G is lower than that of Data A and Data B, which could be because Data F and Data G have a lower average count (Data A: 240, Data B: 490, Data F: 11, Data G:3.7). Overall, this analysis not only helps us understand the reasons for low RR values but also

provides us with greater insight into the RR values obtained. We have added Accuracy and Spread to Table 1 and included these additional explanations in the main text.

We also appreciate the suggestion to perform the analysis similar to Figure 3 for Data C and Data D:

In both datasets, we observe that increasing the number of rounds does not necessarily result in a decrease in RR. This is counterintuitive, as we'd normally anticipate more accurate score estimations with a greater number of rounds. One possible explanation for this behavior is the presence of many variants with high 'volatility', where the corresponding counts fluctuate largely across rounds (e.g. a variant may nearly disappear in one round but show a significant increase in count in the next, or vice versa.). Note that this high 'volatility' is a clear signature of the low Accuracy in these two experiments, as the presence of many variants with high volatility will increase the model-predicted noise. For a variant characterized by such volatility, removing rounds can paradoxically lead to an increase in score errors, as illustrated in the following figure:

We expect that increasing more rounds could stabilize this behavior, as better variants will become more dominant in the library.

Regarding the total NGS reads, for Data-C, there is a very slow increase as the total NGS reads increase. For Data-D, we hardly observe any gain. This is understandable because the total number of variants is less than 1000 for Data-D, so it's nearly saturated in terms of the total NGS reads. If asked for our opinion on how to improve Data-C and Data-D, we would recommend conducting replicate experiments. These replicates would not only help improve the estimation of the score, but they could also enhance the accuracy of alpha and beta estimations. Furthermore, as more data becomes available, we might be able to refine our noise model, specifically tailored to data with volatile behaviors.

3. It would be helpful for the authors to discuss the potential applicability of ACIDES to other types of NGS-based DMS data sets, particularly those generated by FACS-seq.

FACS-seq experiments begin by performing FACS to sort cells into several bins, followed by NGS sequencing on each bin. Scores for the variants are determined based on their estimated abundance in each bin (derived from the sequencing data). ACIDES, however, uses a background framework of the exponential model (Eq. (1) in the manuscript), which does not correspond to this multi-bin setting. Thus, ACIDES cannot be applied to FACS-seq in its current form. Nonetheless, we expect that modifying the background framework appropriately could allow us to use the same negative-binomial noise model and the

log-likelihood maximization method. We defer this interesting line of research to future developments. We have added this explanation in a paragraph of the Discussion section.

Additionally, it is not clear why only a small number of data sets were chosen for benchmarking, given the numerous reported NGS-based DMS data sets available. Please explain why the select set is of sufficient representation.

We have now included the analysis of 12 additional datasets, and used them to benchmark ACIDES against previous methods presented in the literature. Please refer to the new Fig. 5, the second paragraph of "Comparison with previous work" section, and the reply to the following comment 4.1.

4. Benchmarking with Enrich2

4.1. It is not clear why only Data-A and Data-B were utilized for comparing ACIDES and Enrich2, while the performance of ACIDES on Data sets C-G was much worse. The authors implied that the experiments of Data sets C-G were either not well executed or intrinsically more difficult, but it is possible that ACIDES is only effective for certain types of tasks that are similar to those performed for Data-A and -B?

In the previous version of the manuscript, we benchmarked only on data A and B because, among the seven datasets, only these two contained replicates sharing the same initial library, thereby providing a fair basis for comparison. However we agree with the reviewer that this was not sufficient. We now added 12 new datasets that were published after 2017, and compared in detail between ACIDES, enrich2, and another state-of-the-art algorithm DiMSum [Faure et al. 2020]. Following [Faure et al. 2020], we performed leave-one-out cross-validation by using one replicate for test and others for training. The quantity to compare is the statistics of the z-score of enrichments. (Please refer to the manuscript for the details. The key idea here is that the closer the statistics are to the standard normal distribution, the better the method is.) In the following figure, we use the red box plot to compare the inverse values of the standard deviations for the z-score, which is the same quantity used in Fig. 3e in [Faure et al. 2020]. Here, the closer the value to 1, the better the method performs. As can be seen, ACIDES has a clear advantage over Enrich2. while having similar performance to DimSum. In order to deepen this additional benchmark, we also tested for the whole z-score distribution, beyond the simple standard deviation tested in the figure above - see the new methods section for more technical details. In this case (as shown in the blue box plot below) ACIDES outperforms Enrich2 and even does slightly better than DimSum.

We have included this new figure (Fig. 5) in the manuscript and updated the corresponding section, "Comparison with previous work", as well as the tutorial file to explain this comparison between ACIDES and the other algorithms.

4.2. For method evaluation, the authors utilized only the scoring consistency among replicates, but there are more evaluation metrics. For example, it would be valuable to examine how ACIDES and Enrich2 scores and errors correlate with each other, particularly for Data C-G. Additionally, if secondary confirmation data are available in the literature, it would be worthwhile to investigate how ACIDES and Enrich2 ranks correlate with ground truths.

Following the suggestion, we prepared a figure to show how ACIDES and Enrich2 scores and errors correlate with each other.

As can be seen, Data-A, B, E, F and G have correlations almost close to 1, while Data-C and D have lower correlations. The reason why Enrich2 and ACIDES produce similar variant scores for the first group is because they both are based on the assumption of exponentially increasing counts. When there are no volatile variants (please refer to the answer to the comment 2.3), ACIDES and Enrich2 will predict similar scores. The difference appears in the estimated errors, as seen in small correlations in the figure. This is where ACIDES demonstrates its advantage over enrich2. As another reviewer has noted, we did not sufficiently emphasize this point in the previous version of the manuscript, making the description misleading. **We have modified the first paragraph of the “Comparison with previous work” section to enhance the role of the estimated errors in our analysis.**

For Data-C and D, lower correlations in scores are due to the presence of many variants with high ‘volatility’, where the corresponding counts fluctuate largely across rounds (e.g. a variant may nearly disappear in one round but show a significant increase in count in the next, or vice versa.).

In the figure above, we illustrated how ACIDES and enrich2 fit their scores to this count data. As can be seen, the enrich2-fitted line is anchored to the largest and the second largest counts because it employs Poisson errors, which emphasize these points and disregard the other 0 counts. The obtained score is high and this is counterintuitive in the figure, as we would expect disappeared variants to have low scores. Conversely, ACIDES introduces a large dispersion to account for this volatility, so the resulting scores reflect all the points, including 0 counts. Consequently, the score becomes negative, a result that is intuitively justifiable.

As shown in the response to the first comment, we have extensively tested ACIDES and Enrich2 in synthetic datasets where the ground-truth rank is known, and observed that ACIDES outperforms Enrich2.

4.3. Enrich 2 can estimate variant ranks from only one enrichment round (two timepoints). For this type of data, will ACIDES outperform Enrich2 as well?

We thank the referee for this comment. The updated version of ACIDES can now be also applied to datasets with two timepoints. As shown in the new Fig. 5 with 12 additional datasets, ACIDES outperforms Enrich2 and has similar, if not better, performance than DiMSum [Faure et al. 2020]. Please refer to our reply to the comment 4.1 for additional details.

Reviewer #1 (Remarks to the Author):

I thank the authors for their thoughtful incorporation of all the reviewer comments. I have no further concerns or suggestions.

Reviewer #2 (Remarks to the Author):

The authors went to great lengths to act on the suggestions made by me and the other reviewers. I feel that the manuscript is now suitable for being accepted.

Reviewer #3 (Remarks to the Author):

The authors have effectively addressed most concerns from the initial review, resulting in a more coherent paper. However, a remaining issue is that the generally acknowledged concept of directed evolution entails at least two rounds of mutagenesis and selection/screening. The manuscript primarily focuses on mutant enrichment via repeated selection from a single library, which diverges from traditional directed evolution, making the use of the term in the title and manuscript misleading.

Authors' response and summary of changes

Reviewer #1 (Remarks to the Author):

I thank the authors for their thoughtful incorporation of all the reviewer comments. I have no further concerns or suggestions.

We would like to thank the reviewer once again for their positive comments, careful review of our manuscript, and constructive suggestions.

Reviewer #2 (Remarks to the Author):

The authors went to great lengths to act on the suggestions made by me and the other reviewers. I feel that the manuscript is now suitable for being accepted.

We would like to thank the reviewer once again for their positive comments, careful review of our manuscript, and constructive suggestions.

Reviewer #3 (Remarks to the Author):

The authors have effectively addressed most concerns from the initial review, resulting in a more coherent paper. However, a remaining issue is that the generally acknowledged concept of directed evolution entails at least two rounds of mutagenesis and selection/screening. The manuscript primarily focuses on mutant enrichment via repeated selection from a single library, which diverges from traditional directed evolution, making the use of the term in the title and manuscript misleading.

We would like to thank the reviewer once again for their positive comments, careful review of our manuscript, and constructive suggestions.

We understand the concerns raised by the reviewer about the use of the term "directed evolution", so we have revised the title and the manuscript by using the term "forward genetic screens".

The new title is

ACIDES: on-line monitoring of forward genetic screens for protein engineering
and the new abstract is

Forward genetic screens of mutated variants are a versatile strategy for protein engineering and investigation, which has been successfully applied to various studies like directed evolution (DE) and deep mutational scanning (DMS). While next-generation sequencing can track millions of variants during the screening rounds, the vast and noisy nature of the sequencing data impedes the estimation of the performance of individual variants. Here, we propose ACIDES that combines statistical inference and in-silico simulations to improve performance estimation in the library selection process by attributing accurate statistical scores to individual variants. We tested ACIDES first on a novel random-peptide-insertion experiment and then on multiple public datasets from DE and DMS studies. ACIDES allows

experimentalists to reliably estimate variant performance on the fly and can aid protein engineering and research pipelines in a range of applications, including gene therapy.

As for the main text, we made some changes, such as modifying the last sentence of the first paragraph of the INTRODUCTION to: *Both methods are based on forward genetic screens, and the approach presented in this article can be applied to these fundamental techniques, focusing on their common issues and needs.*